# Direct Feedback Alignment for Recurrent Neural Networks

## Abstract

Time series and sequential data are widespread in many real-world environments. However, implementing physical and adaptive dynamical systems remains a challenge. Direct Feedback Alignment (DFA) is a learning algorithm for neural networks that overcomes some of the limits of backpropagation and can be implemented in neuromorphic hardware (e.g., photonic accelerators). Until now, DFA has been investigated mainly for feedforward architectures. We adapt DFA for both "vanilla" and gated recurrent networks. Unlike backpropagation, the update rule of our DFA can be applied in parallel across time steps, thus removing the sequential propagation of errors. We benchmark DFA on 4 datasets for sequence classification tasks. Although backpropagation still achieves a better predictive accuracy, our DFA shows promising results, especially for environments and physical systems where backpropagation is unavailable.

## 1   Introduction

Backpropagation [Rumelhart et al., 1986] is the long-standing algorithm for credit assignment in artificial neural networks. Its efficient implementation in digital computers has supported the surge of machine and deep learning techniques as one of the key advancements in the field of artificial intelligence [LeCun et al., 2015]. However, with a few exceptions [Wright et al., 2022], the adoption of backpropagation-based learning systems is still mainly limited to digital computers and simulations. It is well known that backpropagation cannot be easily implemented and deployed in physical systems [Momeni et al., 2023, Lillicrap et al., 2020]. For example, due to issues like the weight transport where the synaptic weights of the backward circuit need to be constantly synchronized with the synaptic weights of the forward circuit [Lillicrap et al., 2016, Akrout et al., 2019].

Physical deployment of backpropagation is even more challenging in Recurrent Neural Networks (RNNs) [Elman, 1990], where credit assignment must be performed across time. The most used algorithm to date is BackPropagation Through Time (BPTT) [Werbos, 1990], which extends backpropagation to recurrent architectures.

Over time, several backpropagation-free algorithms have been proposed (see Section 2 for a non-exhaustive overview), some of them with the explicit objective of being compatible with the implementation in physical systems or on unconventional hardware (e.g., neuromorphic, optical). We focus on Direct Feedback Alignment (DFA) [Nøkland, 2016], a backpropagation-free algorithm for credit assignment that removes the weight transport issue and also allows parallel computation of the weight update. DFA has already been implemented in nonconventional hardware, especially photonic [Filipovich et al., 2022]. The photonic co-processor introduced in Launay et al. [2020] scales DFA to trillion-parameter random projections.

Submitted to the Second Workshop on Machine Learning with New Compute Paradigms at NeurIPS (MLNCP 2024). Do not distribute.

We briefly review DFA for feedforward networks in Section 3. We propose an extension of DFA tailored to recurrent neural networks. Our approach is able to compute the update of the recurrent parameters in parallel over all the time steps of the input sequence, thus removing one of the major drawbacks of BPTT. In fact, BPTT sends the error signal computed at the end of the input sequence *back in time* to compute the network parameters update. Instead, the update computed by our version of DFA is local at each time step, as it does not rely on the update computed for other time steps. Due to the weight sharing present in RNNs, the local update is eventually aggregated at the end of the input sequence to compute the final update. The aggregation operation includes information from all the time steps, thus enabling learning of temporal dependencies.

We develop DFA for both a "Vanilla" RNN and a Gated Recurrent Unit (GRU) network [Cho et al., 2014, Chung et al., 2014]. We benchmark both architectures against BPTT on four time-series classification datasets and we find that DFA can achieve non-trivial performances in all of the tested datasets but cannot always attain a performance comparable to BPTT. In general, DFA shows strength in datasets with more than 2 classes and in datasets with a limited number of training samples, although BPTT still surpasses its performance. We show that the GRU architecture trained with DFA is able to learn longer temporal correlations than a "Vanilla" RNN.

## 2   Related works

Lillicrap et al. [2016] proposed the Feedback Alignment algorithm (FA) as a biologically plausible gradient-free learning rule for deep learning. The key idea of FA is to project the errors from the last layer of a deep feedforward architecture to the first layer via random projections between consecutive layers. This simple algorithm has shown competitive performance on the MNIST classification task against the commonly used backpropagation algorithm.

Pushing the FA idea to the extreme, Nøkland [2016] proposed DFA, where the error is randomly projected back to each layer with a direct shortcut connection.

Practical applications of DFA to RNNs have been explored in Nakajima et al. [2022]. The authors performed physical deep learning with an optoelectronic recurrent neural network. However, in their pioneering work, they do not explore the DFA algorithm in the context of fully trainable RNNs, since they only provide a proof-of-concept using a reservoir computing model with untrained reservoir connections [Lukoševičius and Jaeger, 2009]. In this paper, we investigate the potential of DFA on fully-trainable RNNs.

Han et al. [2020] investigated a DFA-inspired algorithm for RNNs. However, their version of DFA is restricted and cannot be applied to any recurrent or gated architecture, like our approach. First, they implement an upper triangular modular structure. Second, they use random projections as powers of the same matrix, which effectively resembles an FA algorithm applied to RNNs rather than a DFA algorithm for RNNs. Overall, our approach stems directly from DFA and closely follows its assumptions without requiring any customization, thus remaining more general and targeting any recurrent model.

## 3   DFA for feedforward networks

We first introduce DFA for feedforward neural networks (Figure 1, middle), to prepare the notation and set the stage for its extension to recurrent neural networks. Consider a fully-connected, feedforward neural network with an arbitrary number of $L$ layers (including input and output layers), input size $I$, hidden size $H$ and output size $O$. Each layer $l$ computes its preactivation $a_l$ through a linear projection $a_l = W_l u_l + b_l$, where $W_l \in \mathbb{R}^{H \times I}, \mathbb{R}^{H \times H}, \mathbb{R}^{O \times H}$ is the weight matrix for the input, hidden and output layers, respectively. Similarly, $b_l \in \mathbb{R}^H$, $l < L$ is the bias vector for the input and hidden layer and $b_L \in \mathbb{R}^O$ is the bias vector for the output layer. The input $u_l$ corresponds to the data sample $x$ for the input layer ($u_1 \in \mathbb{R}^I$) and to the output of the previous layer for all other layers ($u_l \in \mathbb{R}^H, l > 1$). The preactivation at each layer is passed through an element-wise nonlinear function $\sigma$ (e.g., hyperbolic tangent) to generate the layer's activation $h_l = \sigma(a_l)$. The output of the network $\hat{y}$ is read out from the last layer: $\hat{y} = h_L$. For each input example $x$, the loss function $J(\hat{y}, y)$ (e.g., cross-entropy or mean-squared error) measures the error between the output and the target prediction $y$ associated with the example $x$.

Updating the last layer's parameters $W_L, b_L$ via gradient descent is straightforward as there is a direct dependency between $\hat{y}$ and the loss function $J$. For the cross-entropy or the mean-squared error loss,

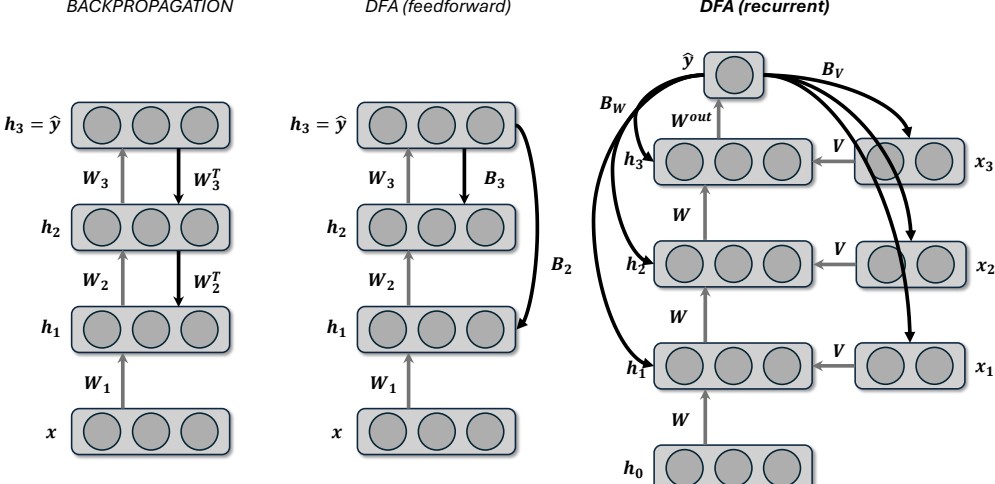

Figure 1: We propose DFA applied to recurrent networks (right). The error is projected through random matrices $B_W$ and $B_V$. We also show backpropagation (left) and DFA (middle) applied to feedforward networks. Grey arrows denote the forward phase, black arrows denote the update phase. Note that in the RNN, the matrices $W$ ad $V$ are shared across time steps (layers), while in feedforward networks each layer has a different matrix. Also, the RNN receives a different input $x_t$ at each time step (here, the input sequence has 3 time steps), while the feedforward network only receives one input $x$.

$e = \frac{\partial J}{\partial a_L} = \hat{y} - y$. Therefore, $e$ can be directly used to update $W_L$: $W_L \leftarrow W_L - \eta e h_{L-1}^T$ and $b_L$: $b_L \leftarrow b_L - \eta e$, where $\eta$ is the learning rate. The update of the last layer's parameters is the same for both backpropagation and DFA.

For the hidden layers, backpropagation computes the update by propagating the error signal $e$ sequentially to lower layers (Figure 1, left). For any hidden layer, we have $W_l \leftarrow W_l - \eta(\ (W_{l+1}^T \delta a_{l+1} \odot \sigma'(a_l))\ u_l^T)$, where $\odot$ denotes element-wise multiplication and $\delta a_{l+1}$ is the error signal coming from *the layer above*. This last term requires the error to be computed sequentially one layer at a time. This dependency prevents updating all layers in parallel.

DFA removes this limitation by projecting the error $e$ *directly* to all layers, through a random matrix $B \in \mathbb{R}^{H \times O}$. $B$ can also be different for each layer. Crucially, the matrix $B$ is kept fixed and only governs the weights update. It does not take any part in the forward phase.

DFA updates each hidden layer via

$$W_l \leftarrow W_l - \eta(\ (Be \odot \sigma'(a_l))\ u_l^T), \tag{1}$$

$$b_l \leftarrow b_l - \eta(\ Be \odot \sigma'(a_l)\ ). \tag{2}$$

These updates can be applied to each layer independently, thus enabling embarrassingly parallel computation for all layers.

DFA also removes the weight alignment issue, as the update circuit uses random connections instead of connections that always need to be synchronized with the forward circuit, like in backpropagation.

## 4 DFA for recurrent networks

We develop a version of DFA that is compatible with RNNs for sequential data processing (Figure 1, right). We closely follow the DFA approach devised for feedforward networks and we extend it to the recurrent case. Each example $x$ is a sequence of $T$ input vectors: $x = (x_1, \ldots, x_T)$, where $x_i \in \mathbb{R}^I$. We consider the sequence classification task where each sequence $x$ is associated with a target class $y$. The RNN keeps an internal hidden state $h \in \mathbb{R}^H$ which is updated at each time step. We first focus on the "Vanilla" RNN [Elman, 1990], whose state update of reads:

$$h_{t+1} = \sigma(Wh_t + Vx_{t+1} + b), \tag{3}$$

where $V \in \mathbb{R}^{H \times I}$ is the input-to-hidden matrix and we call $a_t$ (pre-activations at time $t$) the terms inside $\sigma$. In RNNs, the same layer is applied to all time steps (weight sharing). The output $\hat{y}$ of the RNN is computed from the hidden state: $\hat{y} = \sigma(W^{\text{out}}h_t + b^{\text{out}})$, where $W^{\text{out}} \in \mathbb{R}^{O \times H}$ and $b^{\text{out}} \in \mathbb{R}^O$. The nonlinear function $\sigma$ can be different from the one used in the hidden layers. For sequence classification tasks the output is computed at the end of the input sequence from $h_L$.

Due to the weight sharing, the forward pass of an RNN can be interpreted as the unrolling of the state update function over time. At each time step, the matrix $W$ and $V$ (and the bias as well) are used to compute the next hidden state, much like the matrix $W_l$ is used to compute the layer's output in a feedforward network. The backpropagation algorithm applied to RNNs, called backpropagation through time (BPTT) updates the hidden-to-hidden weight $W$ via $\nabla_W J(\hat{y}, y) = \frac{\partial J}{\partial \hat{y}} \sum_{t=1}^{T} \frac{\partial \hat{y}}{\partial h_t} \frac{\partial h_t}{\partial W}$. The term $\frac{\partial \hat{y}}{\partial h_t}$ hides a dependency between hidden states $\prod_{j=1}^{t-1} \frac{\partial h_{j+1}}{\partial h_j}$ which is due to the sequential propagation of the error over the time steps.

Our DFA-based algorithm for RNN removes this propagation and updates $W$ by computing the term $\frac{\partial J}{\partial \hat{y}} \sum_{t=1}^{T} \frac{\partial h_t}{\partial W}$. The error signal $e$ is projected via a random matrix $B$, randomly initialized and kept fixed.

The equations for the update of $W$ and $V$ via DFA read:

$$W \leftarrow W - \eta \sum_{t=1}^{T} (\, Be \odot \sigma'(a_t) \,) \, h_{t-1}^T, \tag{4}$$

$$V \leftarrow V - \eta \sum_{t=1}^{T} (\, Be \odot \sigma'(a_t) \,) \, x_t^T \tag{5}$$

The bias is updated by omitting the outer product.

**DFA for gated recurrent networks.** In addition to the development of DFA for "Vanilla" RNNs (Equation 3), we also developed a version of DFA for gated recurrent networks, focusing in particular on the GRU network [Cho et al., 2014, Chung et al., 2014]. The state update (forward pass) for a GRU reads:

$$z_{t+1} = \text{sig}(W_z h_t + V_z x_{t+1} + b_z),$$
$$r_{t+1} = \text{sig}(W_r h_t + V_r x_{t+1} + b_r),$$
$$c_{t+1} = \tanh(W_c(h_t \odot r_{t+1}) + V_c x_{t+1} + b_c),$$
$$h_{t+1} = (1 - z_{t+1}) \odot c_{t+1} + z_{t+1} \odot h_t,$$

where *tanh* and *sig* are the hyperbolich tangent and sigmoid functions, respectively. Our DFA update for all parameters of the GRU is provided in Appendix A. The output $\hat{y}$ of the network is computed from the hidden state $h_t$ as previously discussed.

## 5 Experiments

We implemented all our experiments in PyTorch [Paszke et al., 2019]. Although DFA does not compute a true gradient, we filled the "grad" attribute of each weight tensor with the DFA update. This enabled us to use any PyTorch optimizer to apply the update. We used the Adam optimizer for all experiments.

We assessed the performance of DFA against BPTT on the aforementioned "Vanilla" RNN and GRU. We report the average test accuracy and standard deviation computed over 5 runs[1]. Table 1 reports a summary of the time series datasets statistics. We considered 4 different datasets:

---

[1]We will publicly release the code upon paper acceptance.

Table 1: Summary of datasets statistics and average test accuracy and standard deviation over 5 repetitions for all datasets and models.

|  | Strawberry | LIBRAS | ECG200 | Row-MNIST |
|---|---|---|---|---|
| Input size | 1 | 2 | 1 | 28 |
| Number of classes | 2 | 15 | 2 | 10 |
| Sequence length | 235 | 90 | 96 | 28 |
| Dataset size | 983 | 360 | 200 | 70000 |
| DFA GRU | $79.73 \pm 1.23$ | $67.50 \pm 3.68$ | $80.6 \pm 2.25$ | $72.49 \pm 1.1$ |
| BPTT GRU | $92.05 \pm 2.54$ | $80.83 \pm 9.19$ | $82.10 \pm 1.14$ | $99.23 \pm 0.03$ |
| DFA RNN | $67.84 \pm 2.66$ | $47.92 \pm 3.3$ | $78.2 \pm 1.47$ | $87.48 \pm 0.74$ |
| BPTT RNN | $79.08 \pm 4.18$ | $54.30 \pm 18.32$ | $83.30 \pm 2.1$ | $96.69 \pm 0.24$ |

1. *Libras*[2] [Dias Daniel and Helton, 2009] contains 15 classes associated with a different hand movement type. The hand movement is represented as a bi-dimensional curve performed by the hand in a given period of time;

2. *Row-MNIST* [Deng, 2012]: each image of the MNIST dataset is presented to the recurrent model one row at a time;

3. *ECG200* [Olszewski et al., 2001]: where each time series traces the electrical activity of a subject recorded during one heartbeat. The task is a binary classification prediction between a normal heartbeat and one highlighting a Myocardial Infarction;

4. *Strawberry* [K. Kemsley] consists in classifying food spectrographs, a task with applications in food safety and quality assurance. The classes are strawberry (authentic samples) and non-strawberry (adulterated strawberries and other fruits).

The datasets are divided into train, validation and test sets according to the proportions 60%-20%-20%. The hyperparameters have been selected based on a model selection with a grid search (see Appendix B for the details).

Table 1 reports the test accuracy achieved by all methods, alongside the specifics of the datasets. Overall, BPTT still outperforms DFA across most datasets. Specifically, BPTT outperforms DFA with GRU architectures except for the ECG200 dataset, in which both learning algorithms achieve a comparable performance.

With "Vanilla"RNN architectures, BPTT outperforms DFA except for the ECG200 and the Libras datasets, where the average test accuracy of DFA (Figure 2 top-left panel, orange line) is higher than BPTT's one (red line) after the first 150 epochs. Moreover, in this dataset, DFA has the same learning slope of BPTT either with vanilla RNNs (for the first 150 epochs) or for GRUs (for the first 50 Epochs).

DFA seems to struggle with unbalanced datasets, like ECG200 and Strawberry. In the ECG dataset, which is the one with the smallest amount of data, the test accuracy of RNN with DFA is above the random performance of 12%. In the Strawberry dataset, the same model with DFA shows an accuracy which is above the random performance of only 5%. In the case of balanced datasets, RNNs trained with DFA are generally successful at learning temporal correlations.

Overall, while BPTT generally resulted in higher test accuracy, DFA demonstrated comparable performance particularly for ECG200 in both GRU and RNN models. This suggests that although DFA is less accurate overall, it may be a viable alternative in scenarios where strong parallelization combined with a physical implementation is a possibility.

## 6 Conclusion and Future Work

We proposed a learning algorithm for recurrent neural networks based on DFA [Nøkland, 2016]. Our DFA enables parallel updates across the time steps, thus removing the sequential update constraint of

---

[2]LIBRAS is the acronym of the Portuguese name "Lingua BRAsileira de Sinais", is the official Brazilian sign language.

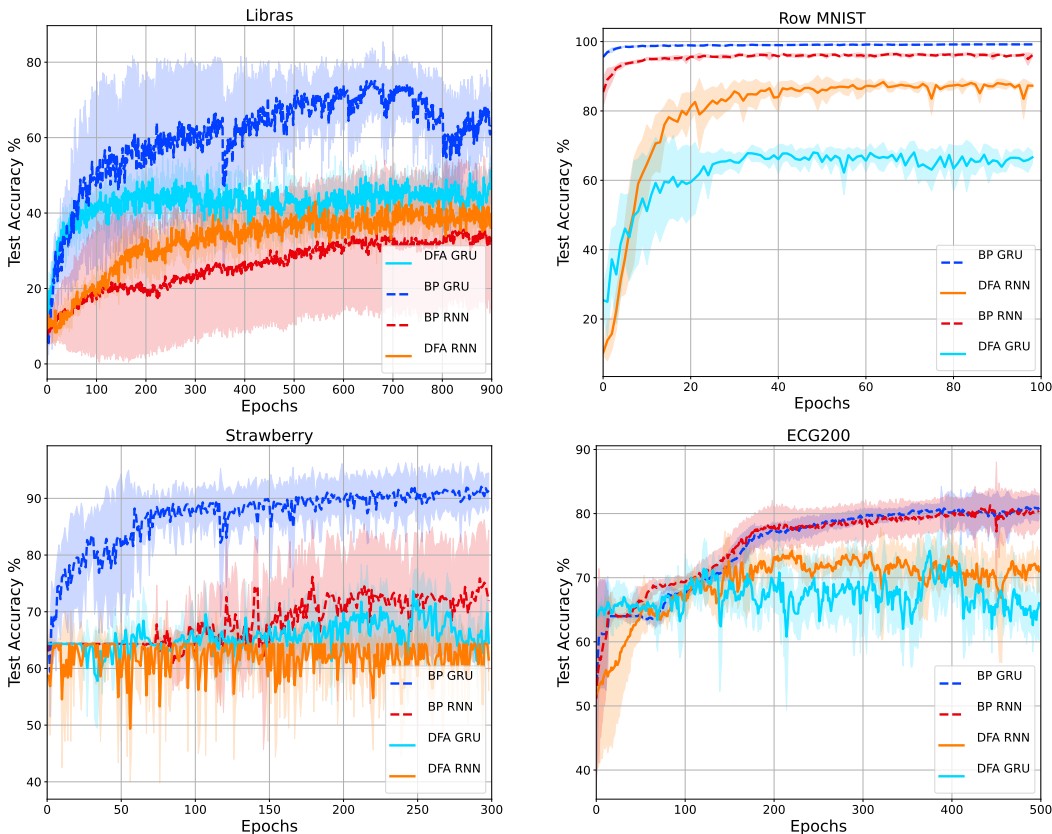

Figure 2: Results on the Libras, Row-MNIST, Strawberry and ECG-200 datasets with a "Vanilla" RNN architecture (orange and red) and with a GRU (blue and cyan). The models are trained with DFA (lighter colors, full line) and BPTT (darker colors, dashed line). Error shades denote one standard deviation computed over 5 repetitions with different seeds.

BPTT. The parallel update phase is particularly interesting for physical implementations of adaptive dynamical systems, as the signal needs not be propagated sequentially back in time. On digital computers, the parallel update allows speed-up when implemented on customized CUDA kernels or with low-level programming interfaces. Unfortunately, in native Python, the speed-up cannot be observed due to the GIL and the large overhead of process spawning. Starting from our publicly available code, future works can refine the implementation, perhaps by integrating the parallel DFA update within the C++ PyTorch API.

There are still other aspects that require further consideration. For example, the choice of the random feedback matrix is crucial, as it affects the trajectory of the parameters during training. Moreover, different matrix structures are amenable to different implementations in neuromorphic or unconventional hardware. Crafton et al. [2019] implemented DFA for feedforward architectures on neuromorphic hardware with a sparse feedback matrix, at minimal or no performance loss. Our algorithm can also be easily extended to deal with time series forecasting tasks, where the prediction step is taken after each time step, instead of only at the end of the input sequence. Further benchmarking of our DFA in these settings is required to understand its effectiveness.

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

# A   Appendix - DFA for Gated Recurrent Unit network

We provide the update rule of DFA for all the parameters of the GRU.

$$W_z \leftarrow W_z - \eta \sum_{t=1}^{T} (Be \odot h_{t-1} - Be \odot c_t) \odot (r_t \odot (1 - r_t)) h_{t-1}^T,$$

$$V_z \leftarrow V_z - \eta \sum_{t=1}^{T} (Be \odot h_{t-1} - Be \odot c_t) \odot (r_t \odot (1 - r_t)) x_t^T,$$

$$W_r \leftarrow W_r - \eta \sum_{t=1}^{T} (W_r(Be \odot (1 - z_t)) * (1 - c_t \odot c_t) h_{t-1}) \odot (r_t \odot (1 - r_t)) h_{t-1}^T,$$

$$V_r \leftarrow V_r - \eta \sum_{t=1}^{T} (W_r(Be \odot (1 - z_t)) * (1 - c_t \odot c_t) h_{t-1}) \odot (r_t \odot (1 - r_t)) x_t^T,$$

$$W_c \leftarrow W_c - \eta \sum_{t=1}^{T} (W_r(Be \odot (1 - z_t)) * (1 - c_t \odot c_t)(r_t \odot h_{t-1})^T,$$

$$V_c \leftarrow V_c - \eta \sum_{t=1}^{T} (W_r(Be \odot (1 - z_t)) * (1 - c_t \odot c_t) x_t^T.$$

As in the "Vanilla" RNN, all the bias vectors are updated by omitting the outer product in the corresponding $W$ or $V$ update. The matrix $B$ can also be a different random matrix for each parameter.

# B   Appendix - Hyperparameter search

Hyperparameters are selected based on the best performances on a validation set among these possible values: hsize$\in$ [50,512], lr$\in$ [0.0005, 0.001,0.005,0.01], bs$\in$ [10,100,256], clip=2. The values selected by the model selection are:

1. Libras: Learning rate = 0.0005 (except for BPTT GRU: learning rate= 0.01), Hidden size = 512, Batch size = 10, Epochs= 900.

2. Strawberry: Learning rate = 0.0005 (except for BPTT GRU: learning rate= 0.005), Hidden size = 50 (except for RNN DFA: hidden size= 512), Batch size = 10 (except for RNN DFA: bs=100 and for RNN BPTT: bs= 256), Epochs= 300.

3. ECG200: [ Learning rate = 0.0005 (Except for DFA GRU, lr=0.01), Hidden size = 50, Batch size = 256, Epochs= 500.

4. ROW-MNIST: [Learning rate=0.0005 (Except for RNN DFA and GRU DFA, lr=0.005), Hidden size = 512 ( Except for RNN BPTT, hs= 50), Batch size = 100 (Except for RNN BPTT, bs = 10)].

In Figure 2 we show the learning curves of the test accuracy for the datasets ECG200 and Strawberry. The fact that the lines start at a different level is because the train, test, and validation sets are divided randomly so the test set can be particularly imbalanced. In these cases, the learning lines of DFA are not visibly growing. We believe that the restricted range of the hyperparameters prevented us to find solutions of DFA that work at best for these datasets.

