# OpenReview forum: "Direct Feedback Alignment for Recurrent Neural Networks"
_NeurIPS.cc/2024/Workshop/MLNCP — Submitted to MLNCP_

### Official Review · Reviewer_wXaW · 2024-09-30
**Relevant topic for the workshop, but the paper is missing key related works, baselines and experiments**

**Rating:** 3
**Confidence:** 5

**Review:**

# Summary
The paper presents a method for training recurrent neural networks (RNNs) without propagating errorsignals back in time through the step-to-step jacobians of the RNN dynamics. The method is based on direct feedback alignment, a popular method that shows promising results on smaller scale feedforward networks. The paper derives update rules for parameters, and it compares the task performance of recurrent networks (vanilla RNN and GRU) trained with this update rule on small scale datasets against training via backpropagation through time (BPTT).
# Discussion
## Missing relevant literature
Training recurrent networks effectively without propagating error signals back in time is a relevant research question for training recurrent networks efficiently on low-resource devices. This question receives significant attention from the neuromorphic computing community (see [1, 2], and there are many others). Gradients can be computed exactly forward in time (i.e. no need for BPTT) with the real-time recurrent learning algorithm [3]. However, this algorithm scales poorly with the model dimensions because it has to carry forward the influence of each parameter on the hidden states of the future. The neuromorphic literature therefore treats the question how this influence can be approximated effectively for learning and efficiently for hardware implementations. Most methods are based on so called eligibility traces that carry forward a low dimensional representation of the aforementioned influence [1]. In fact, setting the eligibility traces to constant identity matrices in [1] is mathematically equivalent to the method proposed in this paper. Since these methods are carefully designed to have only constant compute overhead (e.g. factor of 2 over BPTT), they present relevant baselines, which the present paper should compare against. The paper does not cite or compare against any of these methods, which is a major flaw of this work.

[1] Bellec, Guillaume, et al. "A solution to the learning dilemma for recurrent networks of spiking neurons." _Nature communications_ 11.1 (2020): 3625.

[2] Zucchet, Nicolas, et al. "Online learning of long-range dependencies." _Advances in Neural
Information Processing Systems_ 36 (2023): 10477-10493.

[3] Williams, Ronald J., and David Zipser. "Experimental analysis of the real-time recurrent learning algorithm." _Connection science_ 1.1 (1989): 87-111.

## Missing baselines
It is important to compare against state-of-the-art methods or at least against related works. Not reporting a comparison with related work poses the risk that the baseline presented by the paper (BPTT RNN and BPTT GRU) is poorly tuned, which could potentially unreasonably favor the comparison of the presented method (DFA GRU and DFA RNN) against the baselines. The reviewer suggest to take this into account for future revisions of the paper.

As mentioned above the e-prop algorithm [1] is a highly relevant baseline that combines feedback alignment for the feedforward process with eligibility traces for the recurrent process.

## Evaluation not meeting standards of ML community
The presented method substitutes $\frac{\partial h_t}{\partial h_s}$ for any $t > s$ with the identity matrix for all $t > s$. This substitution ignores all the dynamics of the recurrent networks, and forbids the training procedure to reason about how a state $h_s$ influenced a future state $h_t$. It seems to the reviewer that this would not allow the model to learn arbitrary data distributions well, since the resulting gradient estimator is inherently biased for general data distributions. It is of course a valid and valuable research question to pose if RNNs can be trained under this restriction. It is, however, necessary to benchmark the devised method on a broader range of standardized benchmarks for RNNs. RNNs are (or have been) for example deployed for speech recognition and language processing. Two tasks that arguably require to reason about the relationship between time steps far apart. We know for example that language models improve when trained on longer and longer sequences up to 128k (see LLaMa 3 report for instances). As a proxy for long range reasoning, the Long Range Arena benchmark [4] was proposed. [2] would be a relevant baseline on this task that shows that long range dependencies can in fact be learned by eligibility traces.

The presented benchmarks do not convince the reviewer that the proposed method is able to learn practical tasks well. A recommendation as a first step to make the results more convincing would be to add results when no information is backpropagated in time at all, i.e. removing the recurrent connections from the backward computation (in pytorch this can be realised with `.detach()`e.g. `h_t = torch.tanh(rec_kernel(h_tprevious).detach() + ...)`). To bring this paper to publication a fair comparison to [1] and [2], ideally following the evaluation protocol of [2] is recommended.

It is further standard to show the alignment of the estimated gradient of a novel method with the true gradient via BPTT [5, 6, 2]

[4] Tay, Yi, et al. "Long range arena: A benchmark for efficient transformers." _arXiv preprint arXiv:2011.04006_ (2020).

[5] Tallec, Corentin, and Yann Ollivier. "Unbiased online recurrent optimization." _arXiv preprint arXiv:1702.05043_ (2017).

[6] Mujika, Asier, Florian Meier, and Angelika Steger. "Approximating real-time recurrent learning with random kronecker factors." _Advances in neural information processing systems_ 31 (2018).

## Motivation for erasing the jacobian
Except for the reference to DFA, the paper offers no formal motivation for the proposed method. One such motivation could be formulated as follows: Reparameterize the weight $W = I + \lambda \tilde{W}$ with small $\lambda \ll 1$. In this case, the jacobian $\frac{\partial h_t}{\partial h_{t-1}} = \phi^\prime\left(h_{t-1}\right) \odot \left(I + \lambda \tilde{W}\right)$, which for $\lambda\ll 1$ and an activation function $\phi$ that is approximately identity for $\lvert h_{t-1} \rvert \ll 1$ (such as tanh) boils down to the proposed method when the error is projected back to $h_t$ with the random feedback $B$.
# Comments by line
- 124: missing `\partial` in equation
- 132: typo in hyperbolic

---

### Official Review · Reviewer_eFUX · 2024-09-30
**DFA for RNN**

**Rating:** 6
**Confidence:** 4

**Review:**

Direct feedback alignment (DFA) is a promising alternative training algorithm for BP, since it has advantages of allowing parallel training and easy to implement during the backward process. However, until now,  DFA has been investigated mainly on feedforward neural network. In this study the authors proposed DFA for recurrent neural network (RNN), and utilized 4 benchmark datasets to evaluate proposed method. They proposed that therir method can achieve promising results, thus can be a viable alternative of BPTT when considering physical implementation.
In this study, the methodological derivation part is logical and convincing. However, the experimental results are not clear enough, and more detailed information is needed.  Five are the main points I refer:
1. The results shown in the table are conflict with the fig.2. In the table, for the LIBRAS dataset, BPTT RNN case can achieve 54.30% average accuracy, and DFA RNN case can achieve 47.92%averatge accuracy. But in the fig.2, DFA RNN case outperforms BP RNN case, this is confusing, please modify that.
2. According to the appendix, for the selecting of hyperparameter, the settings of different models under the same dataset shown in table 1 are more or less different, i.e. for the Strawberry dataset, the learing rate if BPTT GRU case is different with others, and the hidden size of DFA RNN also is different with others. Please expain why not use the same settings?
3. In the table 1, DFA RNN case only can achieve 87.48% average accuracy on MNIST dataset, but in the in Nakajima et al. [2022] work, DFA-ESN can achieve 98.96%. Why do fully trained RNN frameworks perform less well than RC frameworks that do not train recurrent weights? Besides, for LIBRAS and Strawberry datasets, BPTT performs unstably, could you please explain this in detail? Or are there existing studies of BPTT on these datasets, and how about their performances? In the row-MNIST dataset, DFA RNN outperforms DFA GRU, but BPTT RNN is worse than BPTT GRU. Please explain this situation in detail.
4. In the table 1, the performance of the proposed DFA is almost significantly lower than that of BPTT in all datasets. Even in the ECG200, there's still a 2% or 5% difference in performance. Can the performance of DFA be further improved through the use of normalization or other techniques to close the gap with BPTT?
5. In the no.65 line, the authors proposed that Han et al. [2020] investigtaed DFA-RNN scheme, but there version DFA is restricted and cannot be applied to any recurrent or gated architecture. Please explain what is their restrictions. Besides, they also applied their approach on GRU and LSTM in their paper, please explain in detail about why their method can not be applied to any gated architecture.

---

### Decision · Program_Chairs · 2024-10-10

Reject